POINT OF VIEW

# Towards a mechanistic foundation of evolutionary theory

**Abstract** Most evolutionary thinking is based on the notion of fitness and related ideas such as fitness landscapes and evolutionary optima. Nevertheless, it is often unclear what fitness actually is, and its meaning often depends on the context. Here we argue that fitness should not be a basal ingredient in verbal or mathematical descriptions of evolution. Instead, we propose that evolutionary birth-death processes, in which individuals give birth and die at ever-changing rates, should be the basis of evolutionary theory, because such processes capture the fundamental events that generate evolutionary dynamics. In evolutionary birth-death processes, fitness is at best a derived quantity, and owing to the potential complexity of such processes, there is no guarantee that there is a simple scalar, such as fitness, that would describe long-term evolutionary outcomes. We discuss how evolutionary birth-death processes can provide useful perspectives on a number of central issues in evolution.

**MICHAEL DOEBELI**[*]**, YAROSLAV ISPOLATOV AND BURT SIMON**

## Introduction

What is evolution? The common definition is that evolution is the temporal dynamics of gene frequencies or phenotype distributions, or generally the change over time of important characteristics of populations of organisms. Such dynamics are thought to occur if there is heritable variation in these characteristics, and if individuals with different characteristics survive and reproduce at different rates, i.e., have different "fitness" (*Lewontin, 1970*). This view of evolution is largely descriptive, and the conditions for evolution to occur have some tautological aspects (evolution occurs when fitter types have more offspring; see, for example, *Ariew and Lewontin, 2004* and *Dawkins, 1982*). This is as if in statistical physics, one would define heat transfer to occur when some objects get warmer and others colder under the condition that there is a difference in temperature between different objects. In statistical physics, a more mechanistic theory instead allows temperature and heat transfer to be seen as emergent properties of the interactions between the

components of matter (such as molecules), and of the complex dynamics that these interactions generate.

In this essay, we advocate a mechanistic approach to evolutionary dynamics that is based on individual organisms undergoing the fundamental events of birth and death. In this approach, the long-term dynamics of genotype or phenotype distributions become an emergent property of the underlying birth-death process. While this line of thought might appear to be straightforward, it leads us to question a number of evolutionary concepts that are held dearly by many. After presenting a general description of evolution as a birth-death process, we discuss some implications of this approach for the concepts of fitness and optimality, for the relationship between ecology and evolution, for social evolution and multi-level selection, and for cultural evolution. Even though in full generality, evolutionary birth-death processes are mathematically intractable, in all these areas a mechanistic model of evolution generates unified perspectives that will be

*For correspondence: doebeli@ zoology.ubc.ca

useful for clarifying empirical questions, as well as for future developments in evolutionary theory.

## Evolution as a stochastic birth-death process

Every model, whether verbal or mathematical, starts with simplifications. From a simplified, yet general perspective, the fundamental processes that give rise to evolution in the biosphere are that individual organisms give birth and die at certain rates (or, equivalently, with certain probabilities per unit time). Individuals have properties, such as genotypes or phenotypes, or age and physiological state, and their birth and death rates depend on these properties. However, in general an individual's birth and death rates also depend on many other things, such as the number and properties of other individuals in the evolving population, properties of individuals in other populations, and the external environment. As a consequence, birth and death rates typically vary in time as populations and the environment change. To obtain a general model of evolution, consider a population of individuals $x_1, \ldots, x_{N(t)}$, where $N(t)$ is the population size at the current time $t$, and $x_i$ is the type of the $i$-th individual. Here "type" is meant to be very general: it could be a simple scalar like body size, or a very complicated quantity like the whole genome of the individual, or a collection of various quantitative phenotypes such as metabolic reaction rates, enzyme efficiencies, and nutrient absorption rates.

We denote by $b_i(t)$ and $d_i(t)$ the birth and death rates of individual $i$ at time $t$. These rates are in general complicated functions of the type of the individual and of the current state of the environment:

$$b_i(t) = b_i(x_i, e_{bio}(t), e_{abio}(t)) \tag{1}$$

$$d_i(t) = d_i(x_i, e_{bio}(t), e_{abio}(t)). \tag{2}$$

Here $e_{bio}(t)$ stands for the state of the biotic environment of individual $i$ at time $t$, e.g., for all the types of the individuals with which the focal individual $i$ has ecological interactions (including individuals from different species). Similarly, $e_{abio}(t)$ stands for the state of the abiotic environment at time $t$, such as the availability of nutrients, the temperature, pH, etc. Thus, in general birth and death rates are functions of the type of an individual, as well as of the current state of the biotic and abiotic environment.

The stochastic process leading to evolution unfolds through birth and death events that occur with probabilities that are proportional to the birth and death rates. Such processes are examples of Markov processes, which are of central importance in the mathematical theory of stochastic processes (*Dawson, 1993*; *Karlin and Taylor, 1975*; *Van Kampen, 1992*). One can envisage such an evolutionary Markov process as a set of clocks that tick, on average, with rates given by the birth and death rates of each individual, with each actual tick drawn from an exponential distribution whose mean is the inverse of the average tick rate (so that clocks with higher rates tick faster). Because birth and death rates depend on the current state of the system, they have to be updated after every event (i.e., after every "tick"), and the resulting stochastic process can be very complicated. Nevertheless, once the birth and death rate functions (*Equations 1* and *2*) are known, evolutionary birth-death processes can usually be simulated efficiently, e.g. using Gillespie algorithms (*Gillespie, 1976*), or other, equivalent stochastic implementations (*Champagnat et al., 2006*, *2008*).

The result is a stochastic process in which the type composition of the evolving population changes as individuals die, in which case their type is removed from the population, or give birth, in which case an individual is added to the population. In the latter case, the type of the newborn individuals is determined by the various processes involved in the production of offspring, such as mate choice, recombination, and mutation. Formally, these processes can be subsumed in an "offspring function" $c(x_i, e_{bio}(t), e_{abio}(t))$, which depends on the type of the individual giving birth, on the biotic environment influencing mate choice, and possibly on the abiotic environment. The function $c$ would also incorporate all the genetic details. Unlike the rate functions $b_i$ and $d_i$, whose values are positive real numbers, the function $c$ takes values in type space. For example, the simplest assumption is that $c(x_i, e_{bio}(t) = x_i$, in which case offspring have the same type as the parent. More generally, for asexual reproduction one can for example assume that $c(x_i, e_{bio}(t)$ is chosen randomly from some distribution with mean $x_i$ and a variance determined by genetic assumptions. With sexual reproduction, the function $c$ becomes more complicated and will incorporate mechanisms such as mate choice and the genetic architecture of traits. In general, the offspring function $c$ contains the "raw material" for

evolution, that is, the genetic processes, such as mutation and recombination, by which novel types are generated.

In this way, the functions $b_i(t)$, $d_i(t)$ and $c(t)$ define a stochastic birth-death process that reflects the fundamental individual-level events that give rise to evolution in the biosphere. One can envisage the resulting dynamics as a cloud of points that moves around in type space. The points in this cloud represent individuals, and the cloud moves because some points disappear (corresponding to death of individuals), while new points appear at new locations in type space (corresponding to birth of individuals that are different from their parents). The collective movement of the cloud represents the evolutionary dynamics, and we think that this description captures the basic structure of evolutionary processes.

In reality, the rate functions $b_i(t)$ and $d_i(t)$ are likely to be exceedingly complex, and even if these functions are not specified, this description reveals a fundamental potential for complexity of evolutionary processes: because the rate functions depend on the biotic environment, individual birth and death rates change as the population composition changes. In stochastic implementations, birth and death rates need to be updated after every single birth and death event, because these events lead to changes in population composition. This generates a basic feedback loop that has the potential to generate complexity in the evolutionary process (this feedback loop is traditionally referred to as frequency dependence, (*Heino et al., 1998*; *Metz and Geritz, 2016*). In addition, the abiotic environment may of course also depend on time, both through biotic effects such as nutrient depletion, and through fluctuations in the physical and chemical environment. Thus, complexity in the rate functions and in the offspring function $c$ can in principle generate complicated evolutionary dynamics (we will give examples of this in the next section).

Even though the description of evolution as a birth-death process is conceptually very simple, the potential for complexity should not really come as a surprise. After all, it seems intuitively obvious that, generally speaking, the biosphere is a complex system (much more complicated than the weather, for example, which is part of the biosphere), and that its composition undergoes complicated temporal changes. It is important to keep in mind that in principle, this complexity of evolutionary dynamics not only comes from novelties generated by mutational processes (which would be described by the function $c$), but also from non-linear feedbacks in birth and death rates.

## Some remarks

Because the above description is based on rate functions, the resulting evolutionary process unfolds in continuous time. It is straightforward to obtain an analogous description in discrete time (e.g. for organisms with non-overlapping generations) by assuming that the functions $b_i(t)$ and $d_i(t)$ are not rates, but probabilities of giving birth and dying over one time step (e.g., one generation). In either case, the basic description of evolution as a stochastic birth-death process can be augmented in many different ways by adding events that impact an organism's life, such as dispersal, or a change in physiological state. These events would again occur at certain rates (or with certain probabilities), with rate functions generally depending on the state of the evolving system. Which events one wants to include in a particular model would depend on the biological questions asked.

Even though births and deaths of reproducing units are a natural starting point for describing evolution as the dynamics of the abundance of different kinds of individuals, the definition of reproductive unit may not always be straightforward. In some fungi or plants, for example, biomass growth may be a reasonable substitute for reproduction, in which case the birth and death rate functions would take on a slightly different meaning and include growth and decline of biomass. The meaning of the offspring function $c$ would have to be adjusted accordingly.

It is also worth pointing out that in the evolutionary birth-death process described, any form of genetic heritability is incorporated in the offspring function $c$. However, formally the birth-death process also represents an evolutionary process with other forms of heritability. For example, the absence of heritability – e.g. if the function $c$ prescribes a random choice in type space independent of the type of the parent – would also result in a form of evolutionary dynamics. More generally, the function $c$ can describe different forms of non-genetic heritability, as can occur for example in cultural transmission and evolution (see below).

## Fitness and optimality

An important, though admittedly self-evident aspect of the perspective of evolution as a birth-death process is that evolution explicitly

becomes a forward-propagating process. What happens next in an evolving system is entirely determined by the current state of the system, and not by any sort of "teleological" quantity. In particular, a priori there is no fitness function that "drives" evolution. Of course, an immediate counterargument to this statement would be that one could define fitness as the difference between birth and death rates. Then the quantity $f_i(t) = b_i(t) - d_i(t)$ could be called the fitness of type $x_i$ at time $t$, and one could argue that types with higher fitness "do better". However, by taking the difference between birth and death rates one obviously loses information, and so a priori it would be better to simply work with separate birth and death rates.

For example, the variance across replicates of an evolving system depends on the sum of $b_i$ and $d_i$. Also, an explicit distinction between birth and death events allows for separate descriptions of the effects of genotypes or phenotypes on fecundity and viability, which gives more flexibility compared to describing these effects based on compound properties such as growth rates. More importantly, as a function of the type $x_i$, the fitness $f_i(t)$ would only be an instantaneous quantity. While $f_i(t)$ would indeed be a measure of "success' of type $x_i$ compared to other types, it would in principle only be so at time $t$, after which the fitness would need to be recalculated, because birth and death rates generally change after each birth and death event. In particular, there is in general no global, time-independent fitness function $F(x_i)$ such that $f_i(t) = F(x_i)$ at all times $t$, simply because in general, the rate functions $b_i(t)$ and $d_i(t)$ are too complicated to allow for the existence of such a function $F$.

This seems obvious, but is worth being emphasized. To this day, mainstream evolutionary thinking is dominated by imagining evolution as a process that maximizes "fitness". This is true both for the theoretical (e.g. population genetic) and the empirical literature, as well as in popular accounts of evolution ("survival of the fittest"). A common metaphor in evolutionary biology is the fitness landscape (*Wright, 1932*). Formally, a fitness landscape is a scalar function on type space whose maxima correspond to "evolutionary optima", i.e., locations in type space that have the highest fitness and are therefore expected outcomes of evolution. Units of such fitness functions are almost never given. When invoking fitness landscapes, most often the implicit assumption is that the fitness landscape is constant in time, or at least constant over time scales relevant for the problem considered. Evolution is then envisioned as an optimization process, with the evolutionary trajectory converging to the (nearest) maximum of the fitness landscape (possibly hindered by factors such as genetic architecture). In particular, in this view evolution is a process that converges to an equilibrium.

In terms of the stochastic birth-death process described above, assuming that evolution is determined by a constant fitness landscape corresponds exactly to assuming that there is a global fitness function $F$ such that $f_i(t) = F(x_i)$ at all times $t$. This would imply that the difference between birth and death rates is constant in time for all types. Clearly, in that case types with the largest difference would prevail over others, and a cloud of points representing an evolving population would, roughly speaking, move to the vicinity of fitness peaks and stay there. However, it is easy to construct examples of evolutionary birth-death processes in which the evolving population does not settle at an equilibrium, but instead continues to move around in type space perpetually. This is true even for single evolving populations (i.e., in the absence of co-evolution between two different species, as e.g. occurs in predator-prey arms races), as illustrated in *Videos 1* and *2*. These videos show movies of clouds of points (i.e., populations) whose evolutionary dynamics unfold in 2- and 3-dimensional phenotype spaces and are driven by competitive interactions. In the examples shown, the evolutionary dynamics are cyclic (*Video 1*) and chaotic (*Video 2*), respectively (see also *Figure 1*).

We have recently argued that in some sense, most evolutionary dynamics should be expected to be complicated if they unfold in a type space that is itself sufficiently complicated (i.e., high-dimensional, [*Doebeli and Ispolatov, 2010a, 2014; Ispolatov et al., 2016*]). This is in stark contrast to the equilibrium perspective resulting from the assumption of a constant fitness landscape. Thus, while scenarios with constant fitness landscapes are a special case of the general birth-death process that describes evolution, such scenarios are not generic, and instead should be the exception rather than the norm. Generally speaking, an evolutionary birth-death process is a complex dynamical system, and there is no reason to expect that there exists a function that can describe the dynamics of such a complex system so that the function increases along trajectories in the state space of the system. In dynamical systems theory, such a

Cyclic evolution

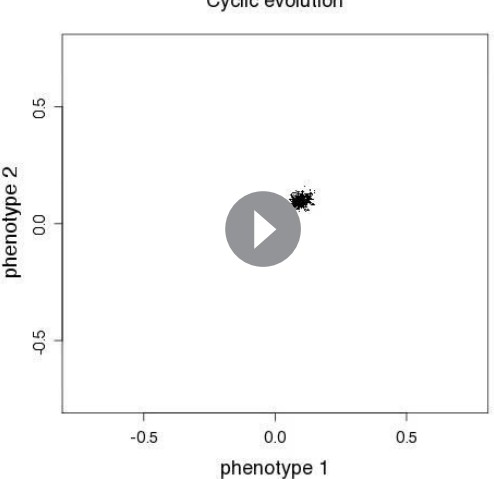

Chaotic evolution

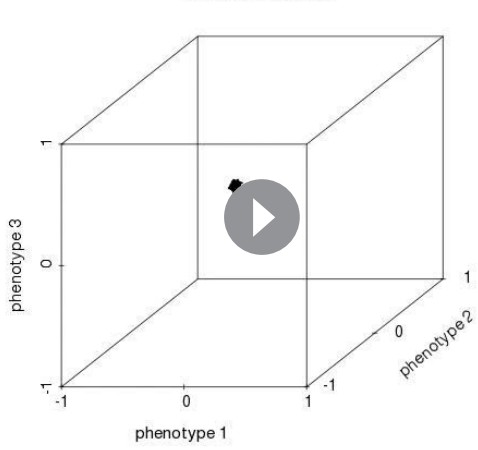

**Video 1.** Evolving populations as clouds of points moving in phenotype space. The points in the cloud represent the individual organisms, which can give birth (spawn a new point in the cloud) and die (the corresponding dot disappears from the cloud). The video shows a single species undergoing cyclic evolutionary dynamics in a 2-dimensional phenotype space. The $x$- and $y$-axis are the two phenotypes, and the cloud of points represents the evolving population. In this example, all individuals had the same constant birth rate, whereas individual death rates were determined by competition from other individuals in the population. Specifically, the death rate of an individual with phenotype $(x, y)$ was equal to $(\sum \alpha(x, y, u, v))/K(x, y)$, where $\alpha(x, y, u, v)$ is the competition kernel, which measures the competitive impact of $(u, v)$ on $(x, y)$, $K(x, y)$ is the carrying capacity of phenotype $(x, y)$, and the sum runs over all other individuals $(u, v)$ in the current population. The video shows the same scenario as Figure 1 in *Ispolatov et al. (2016)*, to which we refer for details of the simulations, as well as of the particular functions chosen for $\alpha$ and $K$.

**Video 2.** A single species undergoing chaotic evolutionary dynamics in a 3-dimensional phenotype space. The cloud of points represents the evolving population. In this example, all individuals had the same constant birth rate, whereas individual death rates were determined by competition from other individuals in the population in a way that is analogous to the example shown in *Video 1*, i.e., based on a particular competition kernel $\alpha$ and a carrying capacity function $K$ (now having 3-dimensional vectors as arguments). The video shows the same scenario as Figure 2 in *Ispolatov et al. (2016)*, to which we refer for details of the simulations, as well as of the particular functions chosen for $\alpha$ and $K$.

function is sometimes called a Lyapunov function (*Devaney, 1986*), and it is well known that most dynamical systems do not have a Lyapunov function, essentially because their dynamics are too complicated. But a Lyapunov function is exactly what a global, time-independent fitness function would amount to: a function in type space that predicts evolutionary trajectories in the sense that the function increases along evolutionary trajectories, so that the eventual outcome of evolution is a maximum of that function. It follows that for general evolutionary birth-death processes, there is no fitness function that would predict their outcome.

In some, and perhaps even in many situations it may be possible to derive such a function from the given birth-death process. In fact, as we argue in the next section, considering certain precisely defined variants of the growth rate $f_i(t)$ can be very helpful in obtaining simplified descriptions of evolutionary birth-death processes. But even in that case it is important to realize that the fitness function would be a derived property of the system, and not a basic ingredient. The basic ingredients of an evolving system are birth and death rates (and the offspring function), and not "fitness". Starting an evolutionary theory based on the notion of fitness thus represents an argumentative leap that is unjustified a priori.

Incidentally, this also applies to evolutionary game theory (*Hofbauer and Sigmund, 1998*; *Maynard Smith, 1982*) and its extensions (*Brown, 2016*), which are generally also based on fitness functions, e.g. in the form of payoff matrices. Evolutionary game theory is a powerful

tool that has generated many important and fundamental insights into processes of frequency-dependent selection, e.g. in the context of social evolution (*Sigmund, 2010*). In particular, evolutionary game theory has greatly expanded the 'fitness horizon' from the notion of constant fitness landscapes. Nevertheless, most game theoretic models have the limitation that they are based on ad hoc fitness functions, which are usually not derived from underlying birth-death processes (for an exception see e.g. *Traulsen et al. (2005)* and *Huang et al. (2015)*, who derive deterministic game dynamics from underlying stochastic processes in finite populations). In principle, to justify the use of fitness in a particular scenario, one needs to first determine the evolutionary birth-death process describing that scenario, and then argue that this evolutionary process admits a fitness function that determines the dynamics of the process. As the field of evolutionary biology stands now, such a justification for the use of fitness is rarely given.

### Towards an analytical theory of evolutionary birth-death processes

Given the rate functions $b_i$ and $d_i$ and the offspring function $c$ for a particular evolutionary scenario, it is straightforward to simulate the birth-death process and to obtain sample trajectories of the resulting evolutionary dynamics. This yields "experimental", i.e., numerical insights, and much can be learned from observing sample trajectories, or from statistics calculated from multiple trajectories. It would of course also be desirable to have analytical descriptions of evolutionary processes, much like in statistical mechanics, whose goal it is to provide an analytical theory of the collective dynamics of a large number of particles. The mathematical theory of stochastic processes is well developed (see e.g. *Karlin and Taylor (1975)* for an introduction), but complete mathematical tractability is only possible for relatively simple birth-death processes. In principle, any evolutionary birth-death process can be described by a differential Master equation for the probability of an evolving system to have a given number of individuals of each type at a given moment in time (*Van Kampen, 1992*). A form of this equation is given in the Appendix. However, such equations can only be solved analytically for simple model scenarios, e.g. when birth and death rates are constant and the offspring function is very simple. In more realistic situations, certain limits and approximations need to be taken, which can lead to

deterministic approximations of the stochastic process in question. For example, Nicholas Champagnat and colleagues (*Champagnat et al., 2006*, *2008*) have shown that under certain conditions, in the limit of large populations evolutionary birth-death processes can be described by partial differential equations. These equations describe the deterministic dynamics of type distributions and are themselves rather complicated objects. Nevertheless, in principle they lend themselves to analytical investigations.

One interesting result is that the type of analytical approximation one obtains depends very much on the type of limits one takes (*Champagnat et al., 2006*; *Dieckmann and Law, 1996*). Thus, different approximations can describe the same evolutionary birth-death process, and the question then becomes which one of these approximations is biologically most relevant. An interesting approximation is obtained by assuming not only that population sizes are large, but also that beneficial mutations are exceedingly rare and small (this is only a rough description, as the details of what "rare" and "small" means are important, see *Champagnat et al. (2006)*, *Dieckmann and Law (1996)*). In that case, the dynamics of the mean type $\bar{x}$ of an evolving population is given by a differential equation

$$\frac{d\bar{x}}{dt} \propto \frac{\partial f(\bar{x}, y)}{\partial y}\big|_{y=\bar{x}}. \tag{3}$$

Here $f(\bar{x}, y)$ is the invasion fitness, which is defined as the per capita growth rate, i.e., the difference between the birth rate and death rates, of an individual with type $y$ in a biotic environment determined by the resident type $\bar{x}$ (in fluctuating resident environments, the mutant per capita growth rate has to be averaged over the relevant time scale of these fluctuations). *Equation 3* is an example of the so-called canonical equation of adaptive dynamics (*Dieckmann and Law, 1996*), which is the theoretical framework for the analysis of evolutionary dynamics derived from invasion fitness functions (*Dieckmann, 2004*; *Geritz et al., 1998*; *Metz et al., 1992*, *1996*). *Figure 1* illustrates the approximation obtained from the adaptive dynamics (*Equation 3*) for the evolutionary birth-death processes shown in *Video 1* and *Video 2*.

Invasion fitness can be derived from evolutionary birth-death processes not only for particular cases, but as a general principle (*Dieckmann and Law, 1996*), and adaptive

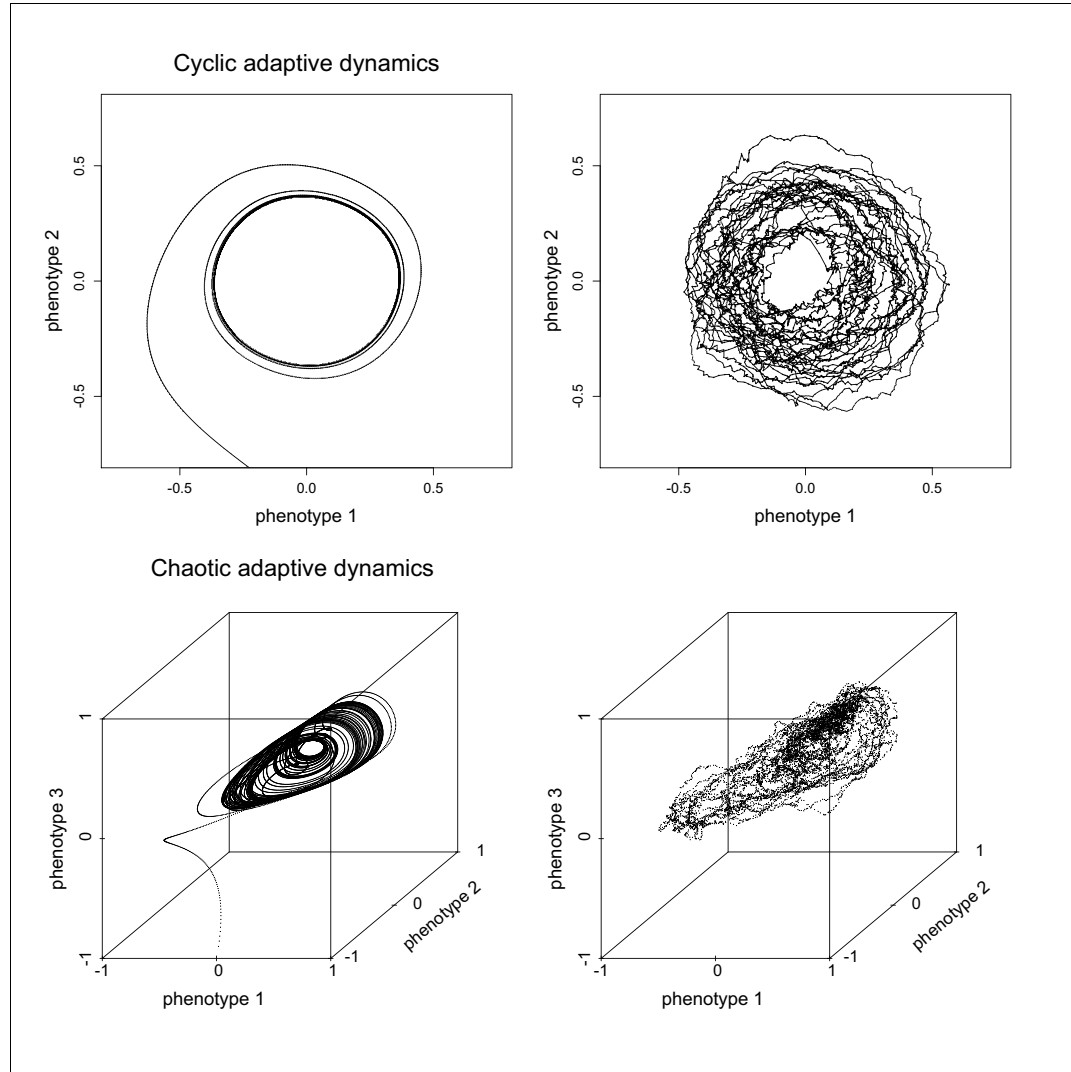

**Figure 1.** Deterministic approximations of evolutionary birth-death processes. Adaptive dynamics, given by *Equation 3*, derived for the two examples of evolutionary birth-death processes shown in *Video 1* and *Video 2*. Adaptive dynamics yields a deterministic approximation of the evolutionary dynamics, which is shown in the left panels (cyclic dynamics in 2-dimensional phenotypes space in the top left panel, chaotic attractor in 3-dimensional phenotype space in the bottom left panel). The panels on the right show the corresponding trajectories of the centres of mass of the clouds of points representing the two evolutionary birth-death processes in *Video 1* and *Video 2*. For details see (*Ispolatov et al., 2016*).

dynamics can therefore be used to gain general insights into such processes. At any given point in time, invasion fitness describes a fitness landscape for all possible mutants appearing in the current biotic and abiotic environment. However, there is a fundamental difference between invasion fitness and the traditional concept of constant fitness landscapes (apart from the fact that the latter are generally not derived from evolutionary birth-death processes). The difference is that the landscape defined by invasion fitness is generally not constant: because the landscape is a function of the current biotic environment, i.e., of the current resident types, it changes as the population evolves. In essence, invasion fitness defines a dynamical system on fitness landscapes. A paradigmatic example of the dynamics of fitness landscapes occurs when populations, even though always evolving 'uphill' on a changing fitness landscape, eventually come to occupy a fitness minimum. This sounds like an oxymoron, but is in fact a generic and robust outcome of the dynamics of fitness landscapes (*Geritz et al., 1998*) (and one that

could of course never occur with constant fitness landscapes). The related phenomenon of evolutionary branching, which is often the consequence of convergence to such fitness minima, has been studied extensively in the context of the important problem of adaptive diversification and speciation (*Dieckmann et al., 2004*; *Doebeli, 2011*).

## Applications

### Relationship between ecology and evolution

Birth-death processes are fundamentally ecological, because they result in temporal fluctuations in the number of individuals in the population. What makes birth-death processes evolutionary is the offspring function $c$, which allows for type innovation in newborns. Clearly, if innovation is prevented by assuming that the function $c$ is the identity, $c(x) = x$ for all types $x$, one obtains a purely ecological birth-death process. From this perspective, the only difference between ecological and evolutionary processes lies in the offspring function $c$, and ecological processes simply become special cases of evolutionary birth-death processes. Formally, there is thus no fundamental difference between ecological and evolutionary birth-death processes, and hence between ecological and evolutionary dynamics.

In practice, the offspring function $c$ may often be close to the identity, reflecting "almost faithful reproduction" (*Metz et al., 1996*), e.g. of quantitative traits such as body size. The difference between ecology and evolution then becomes one of time scales. On shorter time scales, evolutionary innovation is negligible, and hence the birth-death process is mainly ecological, while on longer time scales exploration of larger areas of type space is possible, and evolutionary change becomes important. For example, the concept of invasion fitness discussed in the previous section is based on the distinction between ecological and evolutionary time scales. Invasion fitness is an intrinsically ecological quantity, because it is defined as the (ecological) growth rate of a mutant type occurring in a resident population undergoing purely ecological dynamics. Separation of ecological and evolutionary time scales essentially reflects an approximation of the underlying evolutionary birth-death process, and invasion fitness provides the hinge for connecting ecological to evolutionary dynamics. It must be kept in mind, however, that

separation of ecological and evolutionary time scales may not always be a good approximation of evolutionary birth-death processes, because evolutionary innovations may be large and frequent rather than gradual and rare (e.g. in microbes). In such scenarios, a clear distinction between ecological and evolutionary processes becomes impossible.

### Social evolution and inclusive fitness

Social evolution, and in particular the evolution of cooperation, is one of the fundamental problems in evolutionary biology. There has been much discussion in recent years about how to model social evolution, and about what the correct "fitness measures" are that should be employed in such models (e.g. the paper *Nowak et al. (2010)* elicited a response with 137 signatories). Some researchers have suggested that considering "inclusive fitness", a fitness measure that takes into account an individual's own fitness as well as that of its relatives (*Hamilton, 1964*), is necessary for understanding social evolution, and that inclusive fitness is the quantity that is maximized in Darwinian evolution in general (*Grafen, 2007*, *2014*). The implication would be that there is a global fitness landscape, given by inclusive fitness, that always determines the outcome of social evolution.

However, like any other evolutionary phenomenon, social evolution must be the consequence of some evolutionary birth-death process, and because of their potential complexity (particularly in the context of social interactions), such processes do not in general admit a global fitness function from which their dynamics can be derived. In existing models of social evolution, inclusive fitness is almost never derived from an underlying birth-death process, and deriving an inclusive fitness function capturing the dynamics of the underlying process would only be possible under special assumptions.

In a birth-death process for social evolution, individuals obviously do not "know" anything about maximizing inclusive fitness. Instead, individuals characterized by certain social traits simply give birth and die at certain rates, which are determined independently for each individual. Birth rates (and death rates) of different individuals may of course be correlated, e.g. due to a shared environment, but birth and death rates of each individual are calculated independently, and, in particular, independently of the degree of relatedness of other individuals. Social evolution is then an emergent property of this

process. For example, for the classical problem of the evolution of cooperation it is typically assumed that individuals differ in their contributions to some "public good" that benefits others but whose production is costly. One could e.g. assume that individuals making larger contributions to the public good have a lower birth rate, a higher death rate, or both, and that the presence of the public good in the vicinity of an individual increases that individual's birth rate (or decreases its death rate, or both).

The question then is whether the frequency or abundance of individuals making larger contributions will increase or decrease over evolutionary time. The answer emerges as the birth-death process unfolds, and of course the expectation is that the frequency of cooperative individuals can only increase if on average, their loss (in birth/death rates) due to production of the public good is compensated by gain (in birth/death rates) from the presence of the public good in their vicinity. In some scenarios it may be possible to subsume and describe the emergent dynamics using an inclusive fitness measure, but such a fitness measure would not be a basic ingredient of the underlying birth-death process, and instead could at best be a quantity that is derived from that process. Moreover, if it is possible to derive this quantity, then it is also possible to derive other fitness measures that would describe the dynamics of the process, such as the direct fitness measures used in evolutionary game theory. Direct fitness measures simply calculate the average number of offspring per unit time of the different cooperative types, taking into account the expected cooperative environment of each type, i.e., the degree of positive assortment between cooperating individuals (*Fletcher and Doebeli, 2009*; *Nathanson et al., 2009*).

In reality, many processes of social evolution, e.g. in the microbial world, may be too complicated to readily admit any sort of fitness measure from which their dynamics could be derived. Instead, one would have to resort to determine birth and death rates of individuals as a function of their social traits and their environment, and then simulate the stochastic process (corresponding to what is actually happening in the real world, rather than in the contrived world of fitness models). Thus, the perspective of evolutionary birth-death processes reveals that there is no fitness concept that would be fundamentally necessary for understanding social evolution.

## Multi-level selection

It has been argued that it is often useful to investigate social evolution in the context of group selection, i.e., of multi-level selection scenarios in which fitness accrues to both individuals and groups of individuals (*Wilson and Wilson, 2008*). Somewhat curiously, the main theoretical tool for such investigations is the Price equation (*Price, 1970*). On the one hand, the Price equation is not really a model, but a useful data visualization tool: it takes stock of what happened over one parent-offspring generation (*Price, 1970*; *van Veelen et al., 2012*). On the other hand, this accounting takes place at the level of individual organisms, and hence, again by construction, the Price equation does not describe multi-level processes. Despite this, the Price equation is used to partition what happened over one generation into certain algebraic terms, which are then interpreted in a biological context as terms corresponding to "selection at the individual level" and "selection at the group level". This typically shows that the group selection terms need to have a certain sign for cooperative traits to increase over one generation, hence the necessity of group selection for social evolution. This is questionable terminology, which started a debate about the "equivalence" of group selection and inclusive fitness models, because proponents of inclusive fitness argue that the group selection terms in the Price equation are equivalent to the inclusive fitness terms in the inclusive fitness interpretation of the Price equation.

We think that the perspective of evolutionary birth-death processes can shed light on the issue of group selection, and of multi-level selection in general. If a population of individuals is undergoing an evolutionary birth-death process, and if in this process type A prevails over type B, one could describe this as selection having favoured type A. Further, because individual organisms are the units giving birth and dying in this scenario, one would describe this as selection occurring at the individual level. In other words, in the perspective of evolutionary birth-death processes, individual-level selection requires individual-level events, i.e., individuals undergoing a birth-death process. Generalizing from this, processes of group selection require group level events. For example, groups could fission into smaller groups at certain rates, and groups could go extinct, which would define a birth-death process at the the level of groups. More generally, multi-level selection requires a birth-

death process at multiple levels of biological organization.

In *Simon et al. (2013)* we have introduced evolutionary birth-death processes that unfold at both the individual and the group levels, and we have given examples of social evolution where the social traits are only maintained in the population because of group level events (i.e., if the birth-death process at the group level is turned off, the remaining birth-death process at the individual level leads to the loss of the social traits). We would argue that only in such situations is it appropriate to say that a social trait evolves by group selection (*Simon et al., 2013*). In general, to talk about selection at a certain level of biological organization, one needs to formulate an evolutionary birth-death process at that level. This implies that to define multi-level models of social evolution, one needs to be able to identify units undergoing a birth-death process at the group level (including an appropriate "offspring function" that describes the outcome of group fissioning). This is of course not always possible, e.g. when cooperation evolves in spatially structured populations due to clustering of cooperators, in which case it is often not feasible to identify the clusters as reproductive units due to their ephemeral nature. In that sense, evolution of cooperation due to spatial clustering is not due to group selection, but rather to individual selection in structured environments (leading to assortment of cooperators). In other scenarios, such as the social evolution of insects, it may be more straightforward to define groups as reproductive units, e.g. as colonies of insects. In particular, group selection may be an important driver for the evolution of eusociality (*Nowak et al., 2010*; *Shaffer et al., 2016*; *Wilson and Wilson, 2008*). Group level birth and death events, and hence group selection, may also be very important for the evolution of multicellularity from aggregations of single-celled organisms (*Hammerschmidt et al., 2014*; *Ratcliff et al., 2012*).

### Cultural evolution

Cultural evolution refers to the waxing and waning of ideas and cultural practices throughout the history of human societies. Cultural evolution is a powerful alternative to genetic evolution that shapes a myriad of aspects of human life, from tool making to music and religion (*Boyd and Richerson, 1985*; *Richerson and Boyd, 2005*). Cultural evolution is non-genetic, but is often thought to interact with human genetics, leading to gene-culture coevolution

(*Laland et al., 2010*), in which cultural practices may generate selection pressures for certain genetic changes (such as the ability to adhere to norms (*Chudek and Henrich, 2011*), and genetic predispositions (e.g. in terms of brain function) determine constraints on cultural evolution. That much of cultural evolution must be non-genetic is evident from the enormous cultural and societal changes in the last 100 or 1000 years. Even though cultural evolution is non-genetic, models of cultural evolution nevertheless most often closely resemble population genetic models of human evolution (*Mesoudi et al., 2006*). This is because cultural traits are often identified with the human individuals exhibiting them, and then the dynamics of the abundance of humans with different kinds of cultural traits is described. This would correspond to an evolutionary birth-death process in which the reproductive units are humans, but their type is cultural. The offspring function $c$ would then describe how human offspring inherit culture from their parents, and there may be additional events in the process corresponding to transmission of culture between humans.

An alternative way to model cultural evolution consists of considering cultural ideas as reproductive units themselves, so that the evolutionary birth-death process is defined at the level of cultural content. The initial problem with this approach is to define what exactly a cultural reproductive unit is, but a similar problem of course also exists in traditional models with humans as reproductive units carrying cultural content. Possible examples of cultural units are blueprints for tool making, levels of cooperation, forms of religion, etc. For a representation in the material world, one could identify cultural units with the patterns of neuron firing or memory storage to which they correspond. Once the units are defined, one can envisage them as colonizing human brains, much like microbes colonize and inhabit the human gut. Thus, cultural units live in human habitats (brains), in which they may accumulate or vanish, corresponding to birth and death. For example, an individual brain might acquire a strong belief and later become agnostic again, corresponding to the waxing and waning of a population of cultural units within that brain. Also, cultural units may be transmitted between habitats (brains) through various mechanisms such as conversations, reading, or social media. These transmission events can also be interpreted as birth events, with the offspring function $c$ describing the difference between the original cultural unit

and the unit after transmission (e.g. the difference between a theoretical concept in the teacher and the student). The various events can then be incorporated into an evolutionary birth-death process with cultural units as individuals.

The process unfolds in collections of human brains, and would be analogous to evolutionary birth-death processes for microbiota inhabiting human bodies. One advantage of such a perspective is that cultural evolution would unfold based on what properties of cultural content are good or bad for the propagation of culture itself, rather than based on what is good or bad for humans exhibiting cultural traits. For example, this might give insights into the mechanisms for the spread of cultural practices that are detrimental to humans, such as certain forms of religion, which may spread not because of their effect on humans, but because of their properties as cultural content. For example, a religion may spread because it prescribes its followers to convince at least $n$ other humans, which may be very laborious for the followers, but may accelerate the spread of the religion. Of course, the effects of culture on its habitat, i.e., on humans, would still be of central importance, but cultural evolutionary birth-death processes squarely put the cultural content itself, and not just the humans carrying it, into the centre of attention, thereby allowing for an assessment of adaptive properties of culture from a different perspective.

The approach of considering culture as the reproductive units in an evolutionary process has for example been used in *Mesoudi (2011)* to investigate constraints on cumulative cultural evolution, and in (*Doebeli, 2011*; *Doebeli and Ispolatov, 2010b*) in a model for diversification of religions. Formulating evolutionary birth-death processes in which culture itself constitutes the reproductive units would also allow for multi-level extensions, in which groups of humans hosting a particular cultural content compete with other groups hosting different cultures. Such an extension would consist of incorporating group-level events, as discussed in the previous section and possibly including phenomena such as cultural absorption of one group by another. This could serve to investigate the phenomenon of cultural group selection (*Henrich, 2004*; *Purzycki et al., 2016*).

## Concluding remarks

Organismic evolution in the biosphere is ultimately the consequence of some organisms dying and others giving birth to new organisms.

Therefore, evolutionary birth-death processes are a natural starting point for both verbal and mathematical descriptions of evolution. For understanding the evolution of a type or a set of types, one should start by asking how those types affect birth and death rates, and how those effects depend on the state of the evolving population and its biotic and abiotic environment. The state space of an evolving system is generally complicated and includes not only the space of all possible types (e.g. genotypes or phenotypes), but also all possible states of evolving populations (i.e., type distributions), as well as environmental states. Because birth and death rates in an evolutionary birth-death process are functions of the state of the evolving system, which in turn changes due to to birth and death events, the dynamics of such processes can be very complicated (as illustrated in *Videos 1* and *2*).

This implies that deriving a general analytical theory of evolutionary birth-death processes is difficult, although substantial progress has been made (e.g. *Champagnat et al., 2006*; *2008*, *Dieckmann and Law, 1996*; *Puhalskii and Simon, 2012*; *Simon, 2008*). It also implies that simplified descriptions can only be obtained by making significant assumptions. In particular, finding a "fitness" function that would predict the outcome of the evolutionary process as equilibria located at maxima of the fitness function is in general impossible. In fact, situations in which constant fitness landscapes are a useful metaphor may be exceedingly rare. Once the birth and death rate functions for a given problem are specified, one can of course try to derive a fitness function that describes the process. For example, at any given point in time the current birth and death rates define a fitness landscape, and one could simply make the assumption that this landscape does not change over time scales that are relevant for the evolutionary question at hand. This may be feasible but would require careful argumentation. In any case, it is important to realize that fitness is not a basic ingredient for evolutionary birth-death processes. At best, it is a derived property of the process.

It is interesting to compare evolutionary theory with other theories in the natural sciences. Many empirical theories were introduced in the physical sciences in the 19th century based on a wealth of experimental data and the development of appropriate mathematical tools. Examples include Ohm's law linking electrical current and resistivity, the Navier-Stokes equations

describing flow of liquids and gases, the Mass Action law for chemical kinetics, or the heat conduction equation for the diffusive transport of heat and matter. With advances in understanding the molecular nature of matter in the 20th century, such theories were later "re-derived" from newly discovered microscopic descriptions in a mathematically strict way as a "coarse-grained" limit for the time evolution of average fluxes and rates in macroscopic systems. Such microscopic re-derivations of initially empirical and phenomenological equations allowed to delineate the validity of the coarse-grained theories by revealing the conditions for their applicability, which is usually restricted to systems where the number of particles is sufficiently large, and when systems are not too far from equilibrium.

The history of quantitative descriptions of evolution appears to be different. Notions such as "survival of the fittest", "fitness landscapes" and " evolutionary optima" are used both in narrative descriptions of evolution and in more quantitative models, which in some sense play the same role as coarse-grained equations in physics and chemistry. But due to the formidable experimental difficulties, quantitative laws of evolution based on such notions are difficult to verify empirically. On the other hand, unlike in the case of the molecular nature of the macroscopic laws in physics and chemistry, it has been understood from the very beginning, at least implicitly, that evolution is based on the "microscopic" events of birth and death of individual organisms. Indeed, corresponding microscopic theories have been developed early on, such as Fisher's "fundamental theorem of natural selection" (*Fisher, 1930*), which is essentially a bookkeeping equation based on given birth and death events. Nevertheless, "macroscopic" notions of fitness have continued to dominate both the narrative and the quantitative description of evolution, probably due to their apparent intuitive appeal and obvious mathematical simplicity. The concept of fitness has of course played a major role in the the development of evolutionary theory, and as such has been very useful, and very widely used, albeit often without empirical or "microscopic" justification. As we have argued, such a justification in fact may often be impossible, especially when constant fitness landscapes are invoked.

General conditions under which a fitness measure can be found that is optimized by an underlying evolutionary process have been studied by (*Metz and Geritz, 2016*; *Metz et al., 2008*), whose results confirm that the existence of a constant fitness function that would capture the

evolutionary dynamics requires very special assumptions. Importantly, it is possible to derive a more general, non-constant fitness concept from evolutionary birth-death processes: invasion fitness (*Metz et al., 1992*), obtained from underlying birth-death processes in the limit of large populations sizes and rare mutations, captures not only the fitness landscape at a given point in time, but also the dynamics of the fitness landscape, i.e., how the fitness landscape changes as a result of evolutionary dynamics. Invasion fitness changes as evolution unfolds because it depends on the state of the evolving system, and hence incorporates frequency-dependent evolution. Because it is derived from underlying birth-death processes, invasion fitness is an intrinsically ecological quantity, and it appears to be the only general, i.e., model-independent fitness definition that can provide an explicit link between ecological and evolutionary dynamics.

The perspective of evolution as a birth-death process is straightforward and is applicable to any problem in evolutionary biology. As we have argued, this perspective can shed new light on a number of contentious conceptual issues. For example, it reveals that there is no fundamental distinction between ecological and evolutionary processes, as both are birth-death processes, the only difference lying in the offspring function $c$. Also, this perspective shows that no fitness measure is a basal ingredient of evolution, instead fitness is at best a derived quantity whose applicability is generally restricted and has to be carefully argued in any given situation. In the context of social evolution, the perspective also reveals the nature of multi-level evolution: for selection to occur at multiple levels, birth and death events have to occur at multiple levels, so that evolutionary birth-death processes simultaneously and interdependently unfold at different levels.

As long as reproductive units undergoing birth and death events can be identified, the perspective of evolutionary birth-death processes also applies to non-genetic evolution. In particular, we think that this perspective can yield important insights for cultural evolution, or perhaps even to study evolutionary dynamics in the age of the machines once artificial intelligence has taken over. But traditional genetic and phenotypic evolution remains the core area in which we think this perspective has the most impact. It is based on the first principles of birth and death and views evolution as an emergent phenomenon of the mechanistic dynamics of the elementary particles constituting populations,

i.e., individual organisms. In particular, long-term evolutionary dynamics must ultimately be the consequence of evolutionary birth-death processes, and models linking short-term ecological events to long-term macroevolutionary patterns have recently emerged (*Doebeli and Ispolatov, 2017*; *Gascuel et al., 2015*). The functions determining birth and death rates in evolutionary birth-death processes are generally non-linear and complicated. As a consequence, long-term evolutionary dynamics are often likely to be complicated as well. This is particularly true if evolutionary processes unfold in type spaces that are themselves complicated (e.g., high-dimensional, (*Doebeli and Ispolatov, 2010a*, *2014*, *2017*), which in turn highlights the central problem of understanding the evolution of genotypic and phenotypic complexity. To obtain satisfactory explanations for this and other core problems in evolutionary biology, it will be useful to adopt the perspective of evolutionary birth-death processes.

**Michael Doebeli** is in the Department of Zoology and Department of Mathematics, University of British Columbia, Vancouver, Canada

http://orcid.org/0000-0002-5975-5710

**Yaroslav Ispolatov** is in the Departamento de Fisica, Universidad de Santiago de Chile, Santiago, Chile

http://orcid.org/0000-0002-0201-3396

**Burt Simon** is in the Department of Mathematical and Statistical Sciences, University of Colorado, Denver, United States

*Author contributions:* MD, Conceptualization, Formal analysis, Funding acquisition, Investigation, Visualization, Methodology, Writing—original draft; YI, Conceptualization, Software, Formal analysis, Funding acquisition, Investigation, Methodology, Writing—review and editing; BS, Conceptualization, Formal analysis, Investigation, Methodology, Writing—review and editing

*Competing interests:* MD: Reviewing editor, *eLife*. The other authors declare that no competing interests exist.

## Funding

| Funder | Grant reference number | Author |
| --- | --- | --- |
| Natural Sciences and Engineering Research Council of Canada | 219930 | Michael Doebeli |
| Fondo Nacional de Desarrollo Científico y Tecnológico | 1151524 | Yaroslav Ispolatov |
| John Simon Guggenheim Memorial Foundation | | Michael Doebeli |

The funders had no role in study design, data collection and interpretation, or the decision to submit the work for publication.

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

## Appendix 1

The simplest description of evolution is a stochastic one-step process driven by the intrinsically random and discrete nature of the birth and death events that change the population by one individual at a time. Assuming that the type space is discrete (as is for example the case when types are given by genome sequences), a comprehensive description of such processes is given by a differential Master equation for the probability $P(n_1, n_2, \ldots; t)$ of the evolving system to have certain numbers of individuals $n_i$ of each type $i$ at a given moment in time $t$:

$$\frac{\partial P(n_1, n_2, \ldots; t)}{\partial t} =$$
$$\sum_i \{(n_i + 1) d_i(n_1, n_2, \ldots, n_i + 1, \ldots) P(n_1, n_2, \ldots, n_i + 1, \ldots; t)$$
$$+ \sum_j \sigma_{ij} n_j b_j(n_1, n_2, \ldots, n_i - 1, \ldots) P(n_1, n_2, \ldots, n_i - 1, \ldots; t)$$
$$- n_i [d_i(n_1, n_2, \ldots, n_i, \ldots; t) + b_i(n_1, n_2, \ldots, n_i, \ldots; t)] P(n_1, n_2, \ldots, n_i, \ldots; t)\}$$

(4)

The first two terms account for the gain in the probability of the population state $(n_1, n_2, \ldots, n_i, \ldots)$ due to death of an individual with type $i$, representing the transition from the state $(n_1, n_2, \ldots, n_i + 1, \ldots)$ state to the state $(n_1, n_2, \ldots, n_i, \ldots)$. Similarly, a birth of an individual with phenotype $i$ in the $(n_1, n_2, \ldots, n_i - 1, \ldots)$ state results in the transition from that state to the current state. The reproduction function $c$ is reflected in the coefficients $\sigma_{ij}$, which are the probabilities that an offspring of an individuals with type $j$ has type $i$ (the coefficients are normalized so that $\sum_i \sigma_{ij} = 1$. The third term accounts for the loss in the probability of the current state $(n_1, n_2, \ldots, n_i, \ldots)$ state due to both birth and death processes. In the context of the master equation, the per capita birth and death rates $b_i(n_1, n_2, \ldots, n_i, \ldots; t)$ and $d_i(n_1, n_2, \ldots, n_i, \ldots; t)$ are the birth and death rates, at time $t$, of individuals of type $i$ in a population consisting of $n_1$ individuals of type 1, $n_2$ individuals of type 2, etc. In general the birth and death rates depend on the whole population state of the system, often in a complex way. The sum over $i$ indicates that the change in the probability of the state $(n_1, n_2, \ldots)$ can occur via the change in the number of individuals of any type.

The number of different possible types may be very large, in which case *Equation 4* typically becomes intractable due to the interactions between the types through the birth and death rate functions. It is therefore often more convenient and realistic to consider continuous types in potentially high-dimensional continuous spaces, as well as to consider large population limits. In that case, the discrete index $i$ is replaced by a multidimensional vector $x$, and the population is described by a density distribution $n(x)$, which is a scalar function of the multidimensional argument $x$ (e.g (*Van Kampen, 1992*). The probability distribution $P(n_1, n_2, \ldots; t)$ then becomes a functional $P[n(.); t]$ of the population density $n(x)$ and the master *Equation 4* becomes a functional master equation. For an example of how this equation can be put to use see (*O'Dwyer et al., 2009*), where this procedure as well as the steady-state solutions of the resulting equation are outlined.

While again intractable except for the simplest cases of constant birth and death rates and trivial offspring functions, this equation could be used to compute first and second moments of the population density $n(x)$ and the population autocorrelation functions under certain assumptions about the birth and death rate functions. For appropriate choices of these functions the master *Equation 4* can be Taylor-expanded to second order, producing Fokker-Planck equations which are generally easier to analyze (*Van Kampen, 1992*).

Without going into details, we note that an alternative approach to gaining analytical insights into evolutionary birth-death processes is to view them as examples of a mathematical structure called measure-valued Markov process (*Dawson, 1993*). Such processes can in principle be studied by determining their so-called infinitesimal generator.

> As with Master equations, infinitesimal generators are complicated objects and difficult to derive explicitly unless the birth and death rate functions are relatively simple.

Doebeli *et al.* eLife 2017;6:e23804. DOI: 10.7554/eLife.23804

