## [Decision Letter]

Thank you for submitting your article "Some perspectives on evolutionary theory" to *eLife* for consideration by *eLife*. Your article has been favorably evaluated by Diethard Tautz (Senior Editor) and three reviewers, one of whom, Wenying Shou (Reviewer 1), is a member of our Board of Reviewing Editors. The following individuals involved in review of your submission have agreed to reveal their identity: Paul Rainey (Reviewer 2) and Jeff Gore (Reviewer 3).

The reviewers have discussed the reviews with one another and the Reviewing Editor has drafted this decision to help you prepare a revised submission.

Summary:

All three reviewers support the publication of this opinion piece. Revisions are recommended as detailed below.

*Reviewer #1:*

Doebeli et al. challenges the usefulness of the concept "fitness", and instead advocates a more mechanistic approach based on individual birth and death processes. I share much of authors' views of deriving understanding from first principles (birth and death in this context), although like many others, I also believe that multiple concepts can coexist to shed insights in different ways.

Authors argue that because dynamics of an evolving system are intrinsically complex, the classic analogy of an evolving population "climbing" fitness peaks is often invalid and that the appropriate fitness function may not even exist. Trading "fitness" with an alternative such as "invasion fitness" where changing fitness landscape is considered, we can explain for example evolutionary branching, a phenomenon that does not emerge with constant fitness landscape. Their paper serves as a useful reminder that the suitable degree of abstraction in any model depends on the phenomenon one studies. I do feel however that fitness-related concepts and analogies are still useful in some cases (e.g. using fitness landscape to conceptualize epistasis). This should be mentioned in the text.

With respect to authors' view on the Price's equation, I feel that the Price's equation is a clever and useful tool for conceptualizing social evolution. A phenomenon can often be viewed through different perspectives, as nicely argued in Ben Kerr's chapter "theoretical and experimental approaches to the evolution of altruism and the levels of selection". The usefulness of one perspective does not preclude the usefulness of another perspective. On the other hand, I agree with the authors that multilevel selection can mean different things to different people (e.g. whether there are group-level birth and death processes), and should be clarified.

*Reviewer #2:*

This is an important paper that is highly appropriate for an *eLife* opinion piece. For far too long evolutionary biology has been held back by comforting, but often misleading, representations of underlying process. In advocating a shift away from fitness as a basal descriptor of evolution and toward a birth-death descriptor, Doebeli and colleagues help fuel new ideas and a more complete understanding of the processes by which evolution actually works. Personally, I find the idea that fitness is – at best – an emergent property highly compelling. It makes sense.

Two issues tangential to the authors' main thesis that they may wish to consider. Firstly, a seemingly trivial matter, but one that is not often appreciated and that is the importance of the coupling of death and birth: this coupling ensures that selection works on fecundity and not just on viability. Eliminate the possibility of birth, and selection works purely on viability (and is largely impotent). Add the possibility of birth, and thus the selection works on fecundity, and evolution is potent. It wouldn't take much to make this link.

Secondly, an important shift that comes from considering the primary currency of evolution as birth/death comes from the fact that the focus moves from growth rate. The standard adaptive landscape metaphor, combined with evolution experiments in simple environments, means that fitness is almost always considered a function of growth rate. But in nature this is probably rarely the case. Once consideration is given to criteria for avoidance of death (and thus the possibility of birth) it becomes apparent that selection can work on many properties (and almost certainly does) and in so doing the range of phenotypic states open to evolution vastly increases.

A related issue is that of timescale. Possibly it is beyond the current manuscript – although the authors touch upon this (I think) around section titled 'Some remarks', second paragraph – but time at which death comes about – whether it is with each cell division, or at some other ecologically or evolutionarily defined (or arbitrarily defined) time interval is hugely important. It is when selection comes to work over a time frame that is longer than the doubling time of individual cells that things get interesting.

A further issue warranting comment is the order of birth/death vs. death/birth. At least in models of evolution on graphs involving a birth/death process, the outcome is different depending upon which event comes first. I think it matters. It is the difference between selection rewarding persistence (avoidance of death) vs. selection rewarding growth rate.

In the section on application to multi-level selection the authors might like to consider the work of Hammerschmidt et al. (Nature) 2014 who incorporated into their experimental design an explicit death-birth process precisely as articulated by the authors. The "higher" level, defined by capacity to proceed through a life cycle involving two different ecological stages, being prone to failure (based on the properties of the cells) allowing the possibility that extant types (that avoided extinction) had opportunity to export their reproductive success. Groups competed with groups with the evolutionary process being driven by death/birth events.

Finally, the authors may also want to consider more fully the discomfort that arises from trying to reconcile the stochasticity inherent in death/birth processes with evolution by natural selection. I don't think that are difficult to reconcile, but doing so requires recognition that what we would consider as adaptive phenotypes are underpinned by much more chance (drift) that we are typically comfortable with.

*Reviewer #3:*

I very much like the approach suggested by the authors, as birth/death events are clear and unambiguous. We are all aware that "fitness" is a tricky thing to define, and the approach taken by the authors makes it clear why this is. I especially like the birth/death approach because it greatly clarifies the links between evolution and ecology. In addition, I believe that eco-evolutionary feedback loops will be easier to spot using this approach. I believe that this perspective will be very helpful for the community, and I believe that it could/should be published essentially as is.

I am not familiar with the concept of invasion fitness, and I can't say that the description here was sufficient for me to understand it. For example, how do we average over an infinite time horizon if the invasion fitness is a function of time? (section titled 'Towards an analytical theory of evolutionary birth-death processes', last paragraph).

---

## [Author Response]

*Reviewer #1:*

*Doebeli et al. challenges the usefulness of the concept "fitness", and instead advocates a more mechanistic approach based on individual birth and death processes. I share much of authors' views of deriving understanding from first principles (birth and death in this context), although like many others, I also believe that multiple concepts can coexist to shed insights in different ways.*

*Authors argue that because dynamics of an evolving system are intrinsically complex, the classic analogy of an evolving population "climbing" fitness peaks is often invalid and that the appropriate fitness function may not even exist. Trading "fitness" with an alternative such as "invasion fitness" where changing fitness landscape is considered, we can explain for example evolutionary branching, a phenomenon that does not emerge with constant fitness landscape. Their paper serves as a useful reminder that the suitable degree of abstraction in any model depends on the phenomenon one studies. I do feel however that fitness-related concepts and analogies are still useful in some cases (e.g. using fitness landscape to conceptualize epistasis). This should be mentioned in the text.*

We agree that fitness-related concepts can be useful, but only if they are justified “ecologically”, i.e., based on birth and death events. Otherwise, we think that such concepts are mainly misleading, exactly because they seem useful based on intuitive grounds, but really aren’t if they can’t be derived from underlying mechanisms. We have inserted the following sentence in the (newly named) “Concluding remarks” section:

“The concept of fitness has of course played a major role in the development of evolutionary theory, and as such has been very useful, and very widely used, albeit often without empirical or ‘microscopic’ justification.”

*With respect to authors' view on the Price's equation, I feel that the Price's equation is a clever and useful tool for conceptualizing social evolution. A phenomenon can often be viewed through different perspectives, as nicely argued in Ben Kerr's chapter "theoretical and experimental approaches to the evolution of altruism and the levels of selection". The usefulness of one perspective does not preclude the usefulness of another perspective. On the other hand, I agree with the authors that multilevel selection can mean different things to different people (e.g. whether there are group-level birth and death processes), and should be clarified.*

We have inserted the word “useful” in the corresponding sentence (section titled 'Multi-level selection', first paragraph). We would maintain that the Price equation is not a model, but a particular way of arranging the data coming out of a specified model (or from an experiment). Data visualization can suggest mechanism, but can never prove mechanism. Moreover, to us the Price equation does not seem useful to study group selection, since the number of groups is typically held constant in formulations based on the Price equation, hence there is no birth-death process at the level of groups. Other than that, this reviewer comment did not seem to be an “actionable item”, as we give quite a lot of information about what we would call group selection (based on birth-death process at the level of groups). Please let us know if you would like to see more clarification on how we view group selection (or the Price equation).

*Reviewer #2:*

*This is an important paper that is highly appropriate for an eLife opinion piece. For far too long evolutionary biology has been held back by comforting, but often misleading, representations of underlying process. In advocating a shift away from fitness as a basal descriptor of evolution and toward a birth-death descriptor, Doebeli and colleagues help fuel new ideas and a more complete understanding of the processes by which evolution actually works. Personally, I find the idea that fitness is – at best – an emergent property highly compelling. It makes sense.*

*Two issues tangential to the authors' main thesis that they may wish to consider. Firstly, a seemingly trivial matter, but one that is not often appreciated and that is the importance of the coupling of death and birth: this coupling ensures that selection works on fecundity and not just on viability. Eliminate the possibility of birth, and selection works purely on viability (and is largely impotent). Add the possibility of birth, and thus the selection works on fecundity, and evolution is potent. It wouldn't take much to make this link.*

*Secondly, an important shift that comes from considering the primary currency of evolution as birth/death comes from the fact that the focus moves from growth rate. The standard adaptive landscape metaphor, combined with evolution experiments in simple environments, means that fitness is almost always considered a function of growth rate. But in nature this is probably rarely the case. Once consideration is given to criteria for avoidance of death (and thus the possibility of birth) it becomes apparent that selection can work on many properties (and almost certainly does) and in so doing the range of phenotypic states open to evolution vastly increases.*

*A related issue is that of timescale. Possibly it is beyond the current manuscript – although the authors touch upon this (I think) around section titled'Some remarks', second paragraph – but time at which death comes about – whether it is with each cell division, or at some other ecologically or evolutionarily defined (or arbitrarily defined) time interval is hugely important. It is when selection comes to work over a time frame that is longer than the doubling time of individual cells that things get interesting.*

In response to first two points raised, we have inserted the following passage:

“Also, an explicit distinction between birth and death events allows for separate descriptions of the effects of genotypes or phenotypes on fecundity and viability, which gives more flexibility compared to describing these effects based on compound properties such as growth rates.”

Regarding the time scale issue, we are not entirely sure what the reviewer means, but we think that these concerns are addressed in our approach, at least implicitly: since evolution occurs by propagating a birth-death process, there is no particular time scale any more at which “selection acts”: rather, one can simply chose any time scale of interest and then observe the dynamics of the process over that time scale. (Please also note that on average, the number of birth events must equal the number of death events per unit time, lest the populations either go extinct or explode to infinity.)

*A further issue warranting comment is the order of birth/death vs. death/birth. At least in models of evolution on graphs involving a birth/death process, the outcome is different depending upon which event comes first. I think it matters. It is the difference between selection rewarding persistence (avoidance of death) vs. selection rewarding growth rate.*

This is a rather technical issue (on which we have previously published) that arises in evolutionary models of Moran processes in populations of a fixed, finite size. Such models are not discussed in our paper, and we would prefer to refrain from addressing this technical issue, because we think it does not fit into the present context.

*In the section on application to multi-level selection the authors might like to consider the work of Hammerschmidt et al. (Nature) 2014 who incorporated into their experimental design an explicit death-birth process precisely as articulated by the authors. The "higher" level, defined by capacity to proceed through a life cycle involving two different ecological stages, being prone to failure (based on the properties of the cells) allowing the possibility that extant types (that avoided extinction) had opportunity to export their reproductive success. Groups competed with groups with the evolutionary process being driven by death/birth events.*

Thanks for pointing this out. We have inserted the following sentence:

“Group level birth and death events, and hence group selection, may also be very important for the evolution of multicellularity from aggregations of single-celled organisms.”

*Finally, the authors may also want to consider more fully the discomfort that arises from trying to reconcile the stochasticity inherent in death/birth processes with evolution by natural selection. I don't think that are difficult to reconcile, but doing so requires recognition that what we would consider as adaptive phenotypes are underpinned by much more chance (drift) that we are typically comfortable with.*

Again, this is a valid point, but since we are throughout talking about stochastic processes anyway, we don’t really see how to incorporate this remark. What the reviewer refers to as “natural selection” appears to be based on some sort of deterministic fitness measure, which we argue should not be used unconditionally anyway. In our view, it’s not that one has something deterministic to start with, and then adds stochasticity and asks how the deterministic thing would be affected by that. Rather, to begin with all we have is the stochastic process, from which one can perhaps derive a deterministic theory, which would then have to reflect the effects of stochasticity at least to some extent.

*Reviewer #3:*

*I very much like the approach suggested by the authors, as birth/death events are clear and unambiguous. We are all aware that "fitness" is a tricky thing to define, and the approach taken by the authors makes it clear why this is. I especially like the birth/death approach because it greatly clarifies the links between evolution and ecology. In addition, I believe that eco-evolutionary feedback loops will be easier to spot using this approach. I believe that this perspective will be very helpful for the community, and I believe that it could/should be published essentially as is.*

*I am not familiar with the concept of invasion fitness, and I can't say that the description here was sufficient for me to understand it. For example, how do we average over an infinite time horizon if the invasion fitness is a function of time? (section titled '', last paragraph).*

Due to the nature of our piece, we can’t really go into any technical details, but we have tried to clarify this issue, at least to some extent, in the last paragraph of section titled 'Towards an analytical theory of evolutionary birth-death processes'.